**Data Availability Statement:** All relevant data are within the manuscript and its Supporting Information files.

**Funding:** The author(s) received no specific funding for this work.

# Remanufacturing end-of-life passenger car waste sheet steel into mesh sheet: A sustainability assessment

**Ziyad Tariq Abdullah** [ID] *

Mechanical Techniques School, Institute of Technology Baghdad, Middle Technical University, Baghdad, Iraq

* ziyad_tariq12@mtu.edu.iq

## Abstract

This study analysed the business sustainability of remanufacturing waste steel sheet from the shells of end-of-life vehicles into mesh steel sheet for manufacturing sheet-metal products. Hybrid statistical, fuzzy, and overall sustainability-index curve-fitting models were used to analyse the technical, economic, environmental, management, and social feasibility of remanufacturing, where the sales price, eco-cost savings, and $CO_2$ emission reductions were used as typical statistical indicators. The remanufacturing process was optimised to allocate hardware for a plant recovering 480 $m^2$/shift of waste sheet steel and producing 2851–5520 $m^2$/shift of mesh sheet steel. Six scenarios were used to model the sustainability parameters to normalise the sustainability index values. The sustainability index of each parameter was calculated by multiplying its weight of importance by its weight of satisfaction. The highest sustainability index of 0.95 was calculated for the economic feasibility index, while the lowest sustainability index of 0.4 was calculated for the management feasibility. Remanufacturing of waste sheet steel into mesh sheet steel can be applied with an estimated overall sustainability index of 0.88.

## Introduction

Worldwide, it is estimated that 10 million t of passenger vehicles is scrapped each year [1], which results in the generation of large amounts of waste and high landfill costs. Although extensive recycling processes have been developed to recover the metals, plastics, and other recyclable materials, more sustainable end-of-life strategies are required that can reduce the energy consumption and $CO_2$ emissions of the processing technologies [2]. Most research in the field of recycling of end-of-life vehicles (ELVs) has involved the development of novel disassembly lines, where the hulk of the ELV is moved to different stations to disassemble specific components. A typical productivity target of such disassembly plants is 30,000 vehicles per 250-d year [3, 4]. The direct and indirect costs of the recycling process and the sales price of the final products can be optimised using a mechanised dismantling system, which can double the profit compared to traditional manual dismantling [4]. Higher profits can be achieved by removing selected parts suitable for reuse or upcycling, which also increases the dismantling speed as complete dismantling of all components is not required [2].

**Competing interests:** The authors have declared that no competing interests exist.

Detailed analysis of the energy use and greenhouse-gas emissions related to operation of the recycling plants can produce metrics that are useful for policy makers trying to optimize the management of ELV recycling [5]. Reduction of the cost, energy, and $CO_2$ emissions from such recycling plants can be realised using reverse logistics networks to redesign the process. Modelling of the transportation, storage, and dismantling of ELVs showed that the transportation accounts for about 70% of the total cost of the recycling process, while dismantling only accounts for 25% [6].

To take advantage of the advancements in automotive technology, such as lightweighting and the use of composite materials, passenger cars with old technology and high $CO_2$ emissions should be phased out to increase energy efficiency. The embodied energy of the ELVs can be recovered by recycling and remanufacturing their constituent materials. An ELV is composed of approximately 68% steel, 22% aluminium, and 8% other metals. Hence, there are significant amounts of raw materials with embedded energy that can be recovered with a short economic cycle to help meet the targets of the automotive industry for continually reducing energy use and emissions [6]. The development of innovative materials and manufacturing processes for passenger cars can help increase the efficiency of end-of-life metal recovery by enhancing their ability to be dismantled and recycled. To increase the efficiency of metal separation close to 100%, car manufacturers are being asked to use aluminium or thinner steel sheets to reduce the overall vehicle weight, instead of the use of alternative materials, such as composites, which are more challenging to recycle [5]. In addition, simplifying the design of the metal panels makes them more suitable for remanufacturing. Modern cars that are designed to be recycled at the end of their useful life are more easily disassembled, avoiding mixed waste that is difficult to separate and recycle [7]. A suite of recycling laws have been put in place in Japan, Korea, and China to produce a global recycling supply chain to double the economic benefits, advance dismantling experiments, and promote international cooperation with developing countries [8]. They suggested eliminating heavy dismantling equipment as much as possible to reduce the environmental footprint, and increase productivity with hybrid manual/automated dismantling lines. With the global expansion of electric cars, the use of copper and steel in passenger cars could be increased to enhance the recycling economic chain, which will make recycling a good business. ELVs should be collected in a controlled manner to prevent them from being abandoned or sent to landfill, so that the economic benefits to society can be increased [9]. Another study recommended the use of modified excavator-based dismantling technology to provide a greater degree of control and higher dismantling force, which can improve the separation performance and increase the amount of recovered recyclable materials, although more energy can be consumed compared to manual dismantling [8].

The recycling process includes transportation, storage, and dismantling steps, where the costs of transporting the parts and materials account for up to 70% of the total cost of recycling, while the dismantling costs can exceed 25%. Replacing recycling with remanufacturing requires changes to the facilities, which will result in variations in the transportation costs depending on the relative locations of the processing facilities and dismantling stations. These factors need to be modelled universally to determine the locations of the different kinds of facilities organized within a reverse recycling network and ensure that appropriate facilities are selected or constructed in viable locations [10].

Dismantling of the bodywork of the ELV is sufficient to achieve 85% recycling efficiency, and higher values (above 95%) can be obtained when waste sheet steel (WSS) is also disassembled and recovered [11]. Technological limitations increase the dismantling effort, which limits the achievable recyclability and recoverability of ELV recycling plants. Technological limitations can result in lower values (by 1–3% for recyclability and 3–15% for recoverability),

depending on the boundary conditions. These losses could be minimised by WSS remanufacturing. WSS has a high embodied value, which can be converted into added value by remanufacturing to help increase recyclability and achieve the aim of a circular economy in the automotive industry [12–14].

Recent trends in research related to processing ELVs include: the development of advanced business models of recycling; the development of high-productivity dismantling lines; calculations of the parameters, resources, energy, and emissions of the recycling plants; and the evaluation of sustainable recycling technologies. This study presents an analysis of the viability of a remanufacturing process that converts WSS from the exterior components of ELVs into value-added mesh steel sheet (MSS) as a sustainable end-of-life strategy as an alternative to recycling the recovered steel by smelting. First, the WSS-to-MSS remanufacturing process was analysed based on the potential global waste stream indicated by published data. Then, fuzzy modelling and scenario-based analyses were used to develop numerical indicators. The overall sustainability of the process was divided into the management (M), economic (C), technical (T), environmental (E), social (S), and management (M) sustainability indices. The findings of this study are expected to help researchers, developers, and policy makers decide which sustainability factors are most important when implementing remanufacturing processes.

## WSS–MSS remanufacturing

### Remanufacturing process

Here the WSS–MSS remanufacturing process is described considering the available ELV waste stream and the related cost and environmental factors. The remanufacturing process includes six steps, as shown in Fig 1. First, pre-shredder treatment is used, where exterior steel components such as the bonnet, roof, hood, front and rear doors, and front and rear fenders, are dismantled from the assemblies. Then, the metal sheets are separated from the frames by CNC laser, plasma, flame, or water-jet cutting. The metal is then classified into waste steel to be recycled, and sheets for remanufacturing. Arrays of slots are then cut in the WSS sheets, which are then expanded by stretching the sheet perpendicular to the cutting direction. Finally, the sheet is flattened using a roller to produce the final MSS.

A dismantling line with a versatile flow of the ELVs can achieve complete disassembly of the vehicles with a cycle time that meets the productivity target of converting 30,000 cars per 250-d year into valuable materials [3, 4]. For one dismantling plant, this corresponds to 240,000 m$^2$/year of WSS available for recovery, which could be remanufactured into ~1.43–2.76 million m$^2$/year of MSS. To meet the recycling targets, the authors estimated that 500

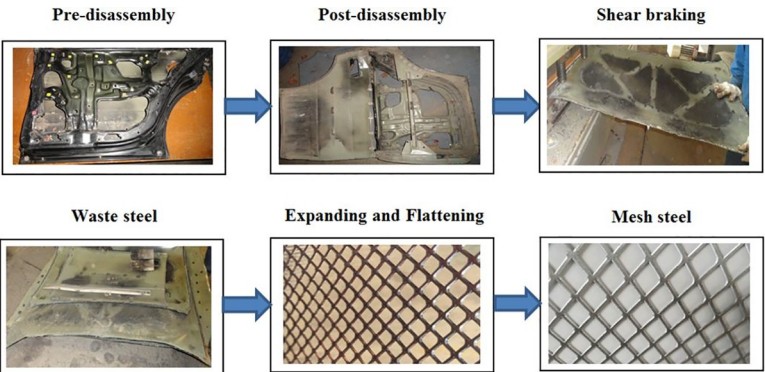

**Fig 1. WSS-MSS remanufacturing.** Flow chart showing the proposed remanufacturing process.

remanufacturing plants will be required, which will introduce 6000 new employment opportunities. Sales of around 7–13.8 million USD/year could be achieved (>60% of which will be profits), while up to ~21,528 t/year of sheet metal products could be produced (saving the equivalent weight of new sheet). In addition, reductions in $CO_2$ emissions of 17–34 million kg/year could be achieved, corresponding to saving 5–10 million USD/year in eco-costs (the cost required to offset the carbon emissions). The energy and $CO_2$ emission reductions from ELV recycling were calculated to be 52.8 MJ/kg and 2.80 $kg_{CO2}$/kg of steel [2]. In the case of remanufacturing WSS into MSS, the expected energy reduction can be increased by around 43–160 MJ/kg or 84–310 MJ/kg, while $CO_2$ emissions could be reduced by a further 3–11 kg $_{CO2}$/kg or 7–21 kg $_{CO2}$/kg, considering the lower and upper bounds of mesh expansion, respectively. Considering a typical thickness of automotive steel sheet of 1 mm, remanufacturing 1 m$^2$ of steel prevents the production of 7.8 kg of new steel, equivalent to a $CO_2$ reduction of 1.559 kg and an eco-cost savings of 0.476 USD/kg. From a total ELV scrap-metal weight of 6.7 million tonne [7], 5.4 million m$^2$ of WSS can be recovered and remanufactured into 318–616 million m$^2$ of MSS, equivalent to 2.5–4.8 million tonne of new sheet steel, which can enhance the economic and environmental outcomes of recycling. This corresponds to a sales price of 1,591–3,082 million USD, a reduction in $CO_2$ emissions of 3.9–7.5 million tonne $CO_2$ (corresponding to an eco-cost saving of 1,182–2,288 million USD).

From a typical passenger car, ~62 kg of sheet steel can be recovered from the eleven exterior components, which can be used to produce ~20–90 m$^2$ of MSS depending on the parameters of the mesh expanding process. Considering the desired productivity of the WSS–MSS process, the following thresholds were defined (per ELV): 6–8 m$^2$ (46.8–62.4 kg) of WSS; 47.52–92 m$^2$ MSS; 370.6–717.6 kg of sheet metal product; 386.8–92.4 kg of used parts; 834.2–818.6 kg of recycled material.

Using a conventional manual disassembly line, a full passenger car can be dissembled in 107 min, with a recovered weight of WSS of ~11–260 kg [3]. In the optimized WSS–MSS process, 1 m$^2$ (7.8 kg) of WSS requires 1 min to be disassembled, and 3 min to be expanded into mesh and flattened. Therefore, the size of the remanufacturing unit, number of workers, and number of machines was determined assuming a laser or water-jet disassembly time of 0.3–1 min, and an expansion time of 1–3 min. Hence, the expansion process is the main bottleneck in the time management optimization problem, followed by the cutting process.

Increasing the weight percentage of steel used in passenger cars from ~62% to ~73% was proposed as an environmentally conscious approach as steel is infinitely recyclable [4]; this would correspond to an increase of ~62 kg (~6%) in the recovered exterior components. For future lightweight vehicles, high-strength steel sheets that are thinner than conventional sheets could be used. The recovered WSS (6–8 m$^2$) can be processed by expanding the sheet to reach the lower remanufacturing target. At least 20 m$^2$ of MSS can be produced from one ELV, which corresponds to 156 kg of new sheet steel. On the other hand, processing to reach the upper remanufacturing target can increase the amount of MSS by 92 m$^2$/vehicle, which corresponds to ~718 kg of new steel.

In 2012, the ELV acquisition cost was ~117–175 USD/t [5], while scrap steel has a market price of ~440 USD/t (2021 price). In 2016, the acquisition price was 75 USD/ELV in China [4]. Hence, WSS–MSS remanufacturing can add value of 108–1000 USD/ELV. Assuming a remanufacturing process portfolio for each facility with a capacity of 60 ELV/d [5] and 30 facilities for dismantling and sorting, to process 25, 50, or 88 ELV/d, 13, 25, or 44 remanufacturing units are required to generate sales of ~0.29–0.56, ~0.56–1.01, and ~0.98–1.89 million USD/d, respectively.

For 1.8 million t of automotive scrap metal, assuming a recycling efficiency of 95% [6], then 1.7 million t of steel is available for recycling, and another 112 t of WSS can be remanufactured

into ~0.7–1.3 million m$^2$ of MSS. This can be used to produce sheet metal products to save the production of 5–10 kt of sheet steel, corresponding to a reduction in $CO_2$ emissions of 8–15 million kg. Thus, a total $CO_2$ emission reduction of ~10.6–18.2 million kg can be obtained, with an additional 2.6 million kg as a result of remanufacturing.

## Comparison of remanufacturing and conventional recycling

The viability of the proposed remanufacturing process is discussed with respect to the conventional recycling process of salvaging spare parts and scrap metal (machine-based dismantling; MBD). The comparison is based on the metrics given in Table 1. The MBD system [15] has an excavator-based multi dismantling machine (MDM), a hydraulic packer for compacting the separated metals into bales, and exhaust fans that run during the operation of the MDM to expel fumes from the hangar. Both systems have a capacity of 125 ELV/d (1220 per 250-d year).

Innovative remanufacturing of WSS can take advantage of MBD to separate the WSS from other waste streams, as shown in Fig 2. To achieve the goal of processing 13 million ELVs to remanufacture their WSS over 10 years, 14 remanufacturing units would be required to process 1.3 million ELV/year at a capacity of 375 ELV/day [8].

The optimization of automotive shredder residue materials produced during conventional recycling can lead to reduction of 21 GJ and 271 kg$_{CO_2}$/ELV in the case of steel [2], while the use of remanufacturing can achieve an energy reduction of 43–160 or 84–310 MJ/kg, and the corresponding $CO_2$ emission reduction would increase by 3–11 or 6–21 kg$_{CO_2}$/kg, assuming the lower and upper remanufacturing limits.

The economic break-even point for recycling ELVs is 30,000 cars per 250-d year [2, 4, 9], corresponding to 120 cars/d. The Pareto front solution of the dismantling cost is 62.4 USD/ELV when 27 ELV/d are dismantled [6]. In this case, 216 m$^2$/d of WSS can be recovered, which can be expanded and flattened into 1283–2484 m$^2$/d of MSS, corresponding to an income of 6415–12,420 USD/d (considering a 20–40% production cost). These factors will need to be considered in optimization models of these processes. Polynomial representation of the Pareto front solution with an error of 3.157% with the introduction of WSS–MSS remanufacturing [4] will change the Pareto solution (62.1 USD/ELV), so that 16 ELV/d satisfies the threshold unit cost of dismantling compared to 27 ELV/d without manufacturing.

The labour cost to dismantle the interior/exterior components is ~430 USD/ELV and the cost of the vehicle is ~58 USD/t in China [4]. However, the direct sale of exterior components

**Table 1. Economic, technical, and environmental comparisons of the WSS–MSS remanufacturing process and machine-based dismantling.**

| | MBD | WSS–MSS |
|---|---|---|
| **Machinery** | MDM powered by a 74 kW diesel engine, operating ~10 h/d, 5 d/week | WSS disassembly machine (20–37 kW) |
| | | Shear braking machine (5.5 kW) |
| | Hydraulic packer (45 kW) | Mesh expanding machine (5.5 kW) |
| | Three 500-mm 0.25-kW exhaust fans | Flattening machine (7.5 kW) |
| **Power consumption (MWh)** | 146.6 | 58.2 |
| **$CO_2$ emissions (t)** | 65.8 | 26.1 |
| **Eco-cost reduction (million USD)** | N/A | 0.2–0.4 |
| **$CO_2$ emission reduction (million kg)** | N/A | 0.7–1.4 |
| **Final product** | Scrap steel | Mesh steel sheet |

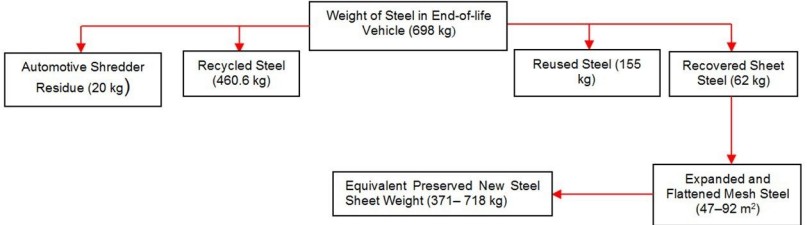

**Fig 2. Breakdown of ELV parts.** Flow chart showing the distribution of the total weight of steel in an end-of-life car between the various waste streams [2].

has a revenue range of ~14–200 USD/ELV, which can be increased dramatically to 237–460 USD by the WSS–MSS remanufacturing and sales of the mesh. The revenue ratio values (MSS price divided by WSS price) for the front and rear doors, boot, hood, and roof are 0.4, 0.4, 0.5, and 0.0, respectively for the case of MBD. In all cases the revenue ratio increases by ~0.5 with the addition of remanufacturing [4].

## Modelling methods

### Calculation of the remanufacturing feasibility indices

The feasibility indices with economic (*C*), technical (*T*), and environmental (*E*) parameters were calculated by first determining an efficiency ($\eta$) that reflects the fraction of added value, as shown in Eq (1), where *VA* is the added value of the remanufactured MSS and *VE* is the embodied value of the WSS. The *VA* and *VE* values were sourced from the literature [7, 16–20], as shown in Tables 3–6. Then, the index (*C*, *T*, or *E*) was calculated using Eq (2) where *W* is the weight of the respective efficiency. The *W* values were sourced from the same studies as the *AV* and *EV* values. The feasibility index values were calculated for three cases with WSS made from: (i) 100% virgin steel; (ii) 100% recycled steel; and (iii) 42% recycled steel. These values were chosen to give a representative range of values. Finally, the feasibility values were plotted and regression consistency tests were used to fit the curves, giving $R^2$ values that were used as the final feasibility values, following the method of [1]. These regression fits are shown in Figs 4–7. The type of relationship (e.g., linear or polynomial) and the order of the polynomial were selected to maximize the $R^2$ value. It is considered that $R^2$ can be used to represent the sustainability index because the regression analysis can be used as a powerful tool for indicating the consistency among the various non-homogenised individual SI indices.

$$\eta_C = (VA - VE)/VA \tag{1}$$

$$\text{Index} = W\eta \tag{2}$$

Statistical data for the reductions in energy, water, and materials use achieved due to the recovery or WSS–MSS remanufacturing process are listed in Tables 2 and 3. The energy-saving efficiency ($\eta_E$), water-saving efficiency ($\eta_W$), and material-saving efficiency ($\eta_M$) values were calculated by substituting the *VE* and *VA* values from Tables 2–4 into Eq (1).

Then, the economic feasibility index (*C*) was calculated using Eq (2) from the *W* and $\eta$ values shown in Table 5 for both remanufacturing ratios. The calculated *C* values were plotted and fit with a fourth-order polynomial, which gave an $R^2$ value of 0.9528 as the final economic feasibility value. Since steel has an infinite ability to be recycled without a loss in quality, recycled sheet steel can be used to manufacture the shell of new cars. This assumption was assumed to define thresholds of *C* = 0.9528.

**Table 2.  VE and VA values for energy, water, and materials usage used for the economic analysis.**

| Region | No. ELVs (million) | Energy (TJ) | | Water (×$10^{12}$ L) | | Material (million m²) | |
| | | VE | VA | VE | VA | VE | VA |
|---|---|---|---|---|---|---|---|
| Global [1] | 100 | 168,480 | 270,578–523,848 | 4.0 | 6.5–12.6 | 800 | 4,752–9,200 |
| Europe [24] | 17 | 28,641.6 | 45,998–89,054 | 0.7 | 1–2.1 | 136 | 808–1564 |
| China [25] | 14 | 23,587.2 | 37,881–73,339 | 0.6 | 0.91–1.8 | 112 | 665–1288 |
| Malaysia [26] | 6.7 | 1,123 | 18,033–34,911 | 0.3 | 0.4–0.8 | 53 | 317–613 |
| Japan [27] | 5 | 8,424 | 13529–26,192 | 0.1 | 0.3–0.6 | 40 | 238–460 |
| Turkey [7] | 3 | 5,054.4 | 8,117–15,715 | 0.09 | 0.19–0.38 | 24 | 143–276 |
| Australia [28] | 0.6 | 1,028 | 1,650–3,195 | 0.02 | 0.04–0.08 | 4 | 29–56 |
| Korea [27] | 0.5 | 876 | 1,407–2,724 | 0.01 | 0.03–0.07 | 4 | 25–48 |
| Belgium [28] | 0.4 | 673.92 | 1,082–2,095 | 0.001 | 0.03–0.05 | 32 | 190–368 |
| Taiwan [27] | 0.27 | 456 | 733–1,419 | 0.007 | 0.02–0.03 | 2 | 13–25 |
| Netherlands [27] | 0.23 | 394 | 633–1,226 | 0.006 | 0.02–0.03 | 2 | 11–22 |

Similarly, the technical feasibility index (*T*) was calculated, where the *AV* and *VE* values were the energy *VE* and *VA* values given in Table 3. The *T* values were plotted and fit with a third-order polynomial, which gave an $R^2$ value of 0.9042 as the final technical feasibility value. Finally, the environmental feasibility index (*E*) was calculated, where the *VA* and *VE* values were kg of $CO_2$ (as shown in Table 4). The *E* values were plotted and fit with a third-order polynomial, which gave an $R^2$ value of 0.8482 as the final environmental feasibility value.

Table 5 shows the *W* values, calculated *η* values, and the corresponding *C*, *T*, and *E* values. Even when only the lower bound is satisfied, the *C* value is considered to be sufficient. Based on the infinite recyclability of steel, thresholds were defined based on the three cases.

## Fuzzy modelling

Fuzzy modelling is a method for representing vague and imprecise information in a logical way. In this study, both criteria and topological discontinuity weak points (TDWP) were used to apply best-worst multi-criteria analysis to determine the sustainability index (SI) of remanufacturing end-of-life passenger car WSS into MSS. Exterior components including the front and rear fenders, front and rear doors, hood, roof, and boot, were used as sub-alternatives to find the most sustainable passenger car design for remanufacturing. A multiple-bottom-line

**Table 3.  VE and VA values for energy (TJ) considering the three different steel source scenarios used for the technical analysis.**

| Region | VE | VA | VA | VA |
| | 100% VSS | 100% recycled | 42% recycled | remanufactured |
|---|---|---|---|---|
| Global [1] | 168,480 | 45,552 | 116,688 | 270,579–523,848 |
| Europe [24] | 28,642 | 7,744 | 19,837 | 45,998–89,054 |
| China [25] | 23,587 | 6,377 | 16,336 | 37,881–73,339 |
| Malaysia [26] | 1,123 | 3,036 | 7,777 | 18,033–34,911 |
| Japan [27] | 8,424 | 2,278 | 5,834 | 13,529–26,192 |
| Turkey [7] | 5,054 | 1,367 | 3,501 | 8,117–15,715 |
| Australia [28] | 1,028 | 278 | 712 | 1,651–3,195 |
| Korea [27] | 876 | 237 | 607 | 1,407–2,724 |
| Belgium [28] | 674 | 182 | 467 | 1,082–2,095 |
| Taiwan [27] | 456 | 124 | 316 | 733–1,419 |
| Netherlands [27] | 394 | 107 | 273 | 633–1,226 |

**Table 4. *VE* and *VA* values for CO$_2$ reductions (million kg of CO$_2$) used for the environmental analysis.**

| Region | *VE* | *VE* | *VE* | *VA* |
|---|---|---|---|---|
| | 100% VSS | 100% recycled | 42% recycled | remanufactured |
| Global [1] | 11,232 | 3,557 | 7,675 | 21,127–40,903 |
| Europe [24] | 1,909 | 604 | 1,304 | 3,591–6,953 |
| China [25] | 1,572 | 497 | 1,074 | 2,957–5,726 |
| Malaysia [26] | 748 | 237 | 511 | 1,408–2,725 |
| Japan [27] | 561 | 177 | 383 | 1,056–2,045 |
| Turkey [7] | 336 | 106 | 230 | 6,33–1,227 |
| Australia [28] | 68 | 21 | 46 | 128–249 |
| Korea [27] | 58 | 18 | 39 | 109–212 |
| Belgium [28] | 4 | 14 | 30 | 84–163 |
| Taiwan [27] | 30 | 9 | 20 | 57–110 |
| Netherlands [27] | 26 | 8 | 17 | 49–95 |

sustainability weighted sum method was used to regulate the relationship among the technical, economic, environmental, social, and management feasibility indices, where the weight of each index was specified based on its priority within the scenario-based analysis, both social and management feasibility indices were estimated by behaviour analysis of the main SI variation curve.

Here, 20 of the best-selling car models produced over the ten-year period of 2009–2019 (i.e., 200 different designs) were studied considering the viability of remanufacturing the WSS into MSS. The designs of the various cars were incorporated into a database and then compared using a fuzzy analysis procedure to determine their SI for the proposed remanufacturing process. Fuzzy modelling has been used successfully in a variety of contexts to apply experience-based analysis [21–23].

The mathematical membership function is:

$$\mu_{\tilde{O}} : R \rightarrow [0, 1] \tag{3}$$

As defined for the triangular fuzzy numbers (TFNs), *l*, *m*, and *u* refer to the lower, medium, and upper values, described by Eqs (4) and (5).

$$\tilde{O} = (l, m, u) \tag{4}$$

$$\mu_{\tilde{O}}(x) = \begin{cases} \dfrac{x - l}{m - l}, & \text{if } l \leq x \leq m \\ \dfrac{u - x}{u - m}, & \text{if } m \leq x \leq u \\ \quad 0, & Otherwise \end{cases} \tag{5}$$

**Table 5. Weight, efficiency, and calculated index values for the three different steel recycling cases.**

| | Economic feasibility | | | Technical feasibility | | | Environmental feasibility | | |
|---|---|---|---|---|---|---|---|---|---|
| | *W* | *η* | *C* | *W* | *η* | *T* | *W* | *η* | *E* |
| Case (i) | 0.2 | 0.377 | 0.075 | 0.2 | 0.377 | 0.075 | 0.2 | 0.468 | 0.094 |
| | | 0.678 | 0.136 | | 0.678 | 0.136 | | 0.725 | 0.145 |
| Case (ii) | 0.3 | 0.377 | 0.113 | 0.5 | 0.832 | 0.416 | 0.5 | 0.832 | 0.416 |
| | | 0.678 | 0.203 | | 0.913 | 0.457 | | 0.913 | 0.457 |
| Case (iii) | 0.5 | 0.377 | 0.189 | 0.3 | 0.569 | 0.171 | 0.3 | 0.637 | 0.191 |
| | | 0.678 | 0.340 | | 0.777 | 0.233 | | 0.812 | 0.244 |

**Table 6. Linguistic terms used in the FFBW sustainability analysis.**

| Linguistic Term | Triangular Fuzzy Number | Reciprocal Triangular Fuzzy Number |
|---|---|---|
| Critically important | (7,9,9) | (1/9,1/9,1/7) |
| Very strongly important | (5,7,9) | (1/9,1/7,1/5) |
| Strongly important | (3,5,7) | (1/7,1/5,1/3) |
| Weakly important | (1,3,5) | (1/5,1/3,1) |
| Equally important | (1,1,3) | (1/3,1,1) |
| Not important | (1,1,1) | (1,1,1) |

The full fuzzy best–worst multi-criterion (FFBW) analysis technique was used study the sustainability performance of the WSS–MSS process with the following algorithm steps:

1. Define a hybrid system of criteria, alternatives, and sub-alternatives

2. Define the importance of each criterion based on published data

3. Pairwise fuzzy reference comparison between the best (worst) criterion and the others

4. Calculate the optimum fuzzy weights to minimize the maximum absolute value by transforming the mathematical model into a fully fuzzy linear programming problem with triangular fuzzy numbers using the linear ranking function

5. Develop a computation model to calculate SI

To evaluate the performance of the FFBW method for analysing the SI, linguistic terms with a triangular fuzzy scale were used for pairwise comparison, as shown in Table 6 [21–23]. The criteria included in the sustainability study are shown in Table 7, along with the results of experience-based analysis regarding the relative importance of the criteria. Those with the largest effect on remanufacturability (and hence, sustainability) are referred to as "best", while those with the smallest effect are referred to as the "worst". The "ease of laser or water-jet disassembly" was defined as the best criteria, while the "ease of flattening" was the worst. Similarly, the results of the analysis of the WSS alternatives are shown in Table 8, where the roof was the best and the front fender was the worst criterion. To calculate SI, the feasibility indices were plotted and their regression values were used to represent the various values of sustainability.

## Criterion constraints

Literature-based and experience-based analysis, and Fig 1 (which represents the experimental work of the study), were used to analyse the criteria, sheet alternatives, and TDWP defects and

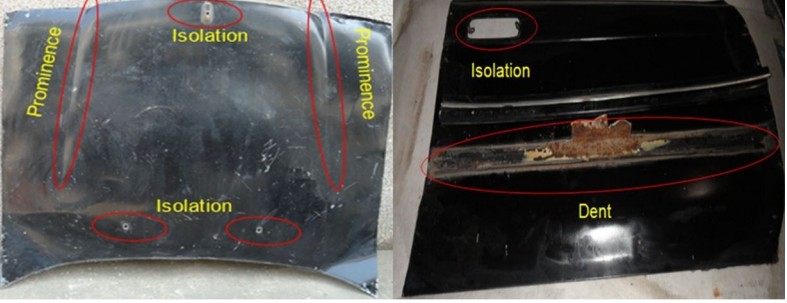

**Fig 3. Definition of topological discontinuities.** Photographs showing representative topological discontinuities that are commonly found in sheets from ELVs.

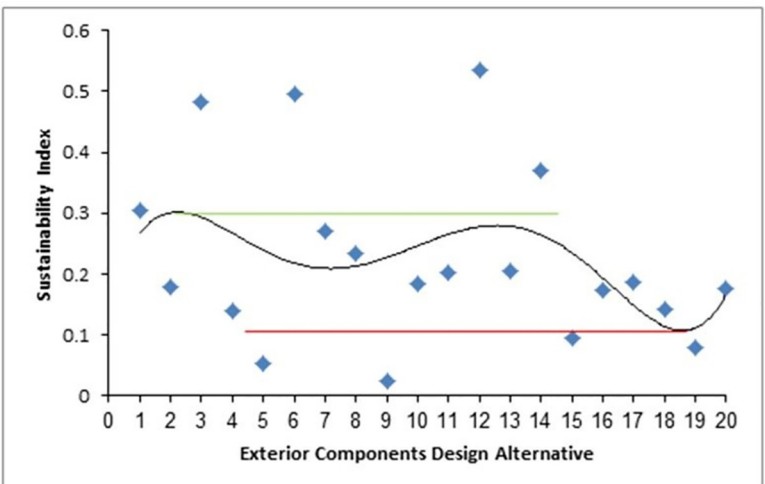

**Fig 4. Sustainability index.** Estimation of sustainability performance variation of WSS from different passenger car designs.

specify their constraints. The potential amount of material that can be recovered from an ELV for WSS–MSS remanufacturing is shown in Fig 2, where the embodied value can be converted into high added value. The criterion constraints are shown in Table 9, where the weight ranges and their ranks on the best and worst scales are listed. The car part alternative constraints are shown in Table 10. A high rank on the best scale indicates that the part has a high embodied value for remanufacturing into mesh.

The sustainability assessment also included a topological discontinuity analysis, as TDWPs in the metal sheets introduced during manufacturing reduce the efficiency of remanufacturing the WSS and limit the technical feasibility. The TDWPs limit the recoverable area of continuous WSS and can cause deformation problems during expansion. TDWPs include prominence, protrusion, isolation, pocket, groove, and dent features (Fig 3). Prominences and protrusions are defects in the metal sheet with a high convexity radius and appear as concave

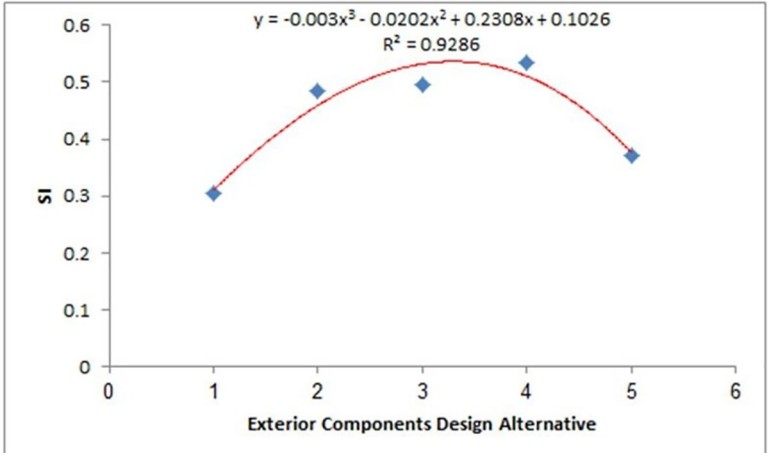

**Fig 5. SI determination (upper).** Fitting of the SI variation of WSS from different passenger car designs with SI values in the upper region of Fig 4.

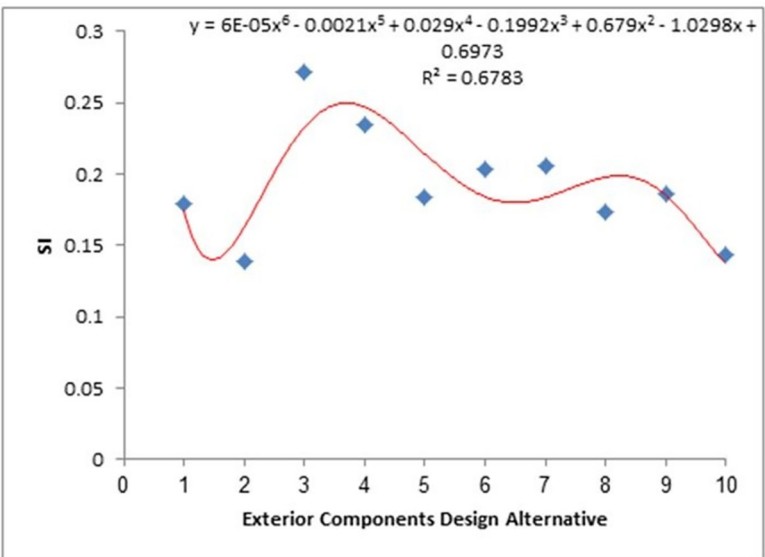

**Fig 6. SI determination (middle).** Fitting of the SI variation of WSS from different passenger car designs with SI values in the middle region of Fig 4.

and convex structures, respectively, when viewed from the outer surface of the sheet. Isolations are non-steel parts embedded in the exterior panel, such as a sun-roof, windscreen washer jet, brake light, roof luggage rack, or antenna. Pockets, grooves, and dents are 3D forming effects with a square, linear concave, and two-sided concave shape, respectively. Isolations, pockets, grooves, and dents are all introduced during the manufacturing phase of the vehicle. The pre-weighting assessment was applied considering the following criteria: (i) intensity of protrusions; (ii) intensity of discontinuities due to non-metal components; and (iii) intensity of cavities and grooves. The TDWP constraints are shown in Table 11. The prominence, dent, and isolation defects had the highest effect on the remanufacturability and aided sustainability, while protrusion pockets and grooves had a smaller effect.

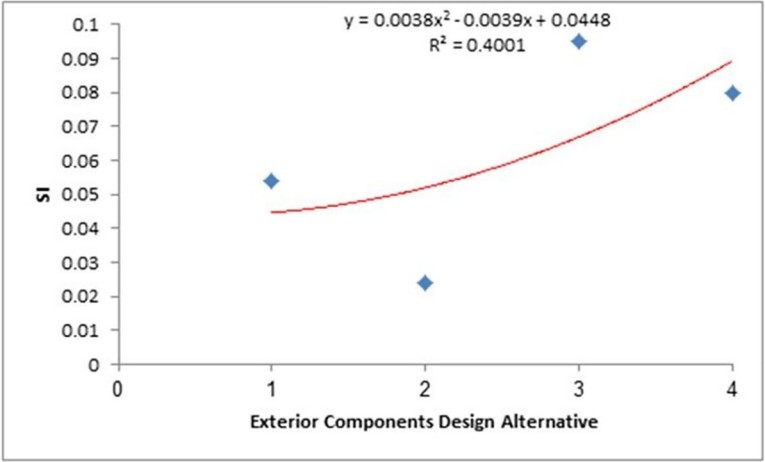

**Fig 7. SI determination (lower).** Fitting of the SI variation of WSS from different passenger car designs with SI values in the lower region of Fig 4.

**Table 7. Comparison of the best criterion "ease of disassembly" and the worst criterion "ease of flattening" with all other criteria.**

| Criterion | Best-to-others vector | Others-to-worst vector |
|---|---|---|
| Ease of disassembly | (1,1,1) | (7,9,9) |
| Ease of mesh expanding | (1,1,3) | (5,7,9) |
| Ease of flattening | (7,9,9) | (1,1,1) |
| Cost saving feasibility | (1,3,5) | (3,5,7) |
| Sheet steel/weight ratio | (1,3,5) | (3,5,7) |
| Sheet steel/mesh steel ratio | (1,3,5) | (3,5,7) |
| Weak points genetic transformation | (1,1,3) | (5,7,9) |
| Eco-cost saving | (5,7,9) | (1,1,3) |
| Pollution reduction feasibility | (5,7,9) | (1,1,3) |
| Employment | (3,5,7) | (1,3,5) |
| Human development | (3,5,7) | (1,3,5) |
| Ergonomic | (5,7,9) | (1,1,3) |
| Workers health | (5,7,9) | (1,1,3) |

## Results and discussion

The SI values for the WSS–MSS process calculated via fuzzy analytical hierarchy modelling are discussed here. The sustainability performance varied according to the various values of SI, which are plotted for the various car designs, where 1 refers to the best-selling car, 2 refers to the second best, and so on (Fig 4). According to Fig 4, the SI variation curve can be divided into three zones to define groups of SI values according to the designated upper and lower thresholds (green and red lines in Fig 4, respectively) to maximise the SI by increasing the SI values during the developmental stage. The upper zone includes the range SI = 0.304–0.535, which represent the threshold to develop the exterior components in a more sustainable way to allow them to be remanufactured more easily. The middle zone was defined as the range of SI = 0.139–0.271, which corresponds to the need for extensive efforts to exceed the development thresholds. The lower zone was defined as the range SI = 0.024–0.095, which includes four unsustainable designs of exterior components, mainly due to the complex curvatures of the panels, which should be avoided to enhance sustainability.

Linear and polynomial regression curves were applied to give $R^2$ values to test the consistency of weights that were given to sustainability criteria and design alternatives. High consistency was observed among the SI values of the upper SI values, indicating a successful assessment process, as shown in Fig 5. Similar plots for the middle and lower SI values are shown in Figs 6 and 7, respectively.

**Table 8. Comparison of the best component "roof" and the worst component "front fender" with all other components.**

| Component | Best-to-others vector | Others-to-worst vector |
|---|---|---|
| Roof | (1,1,1) | (7,9,9) |
| Hood | (1,1,3) | (5,7,9) |
| Boot | (1,3,5) | (3,5,7) |
| Front Door | (3,5,7) | (1,3,5) |
| Rear Door | (3,5,7) | (1,1,3) |
| Front Fender | (7,9,9) | (1,1,1) |
| Rear Fender | (5,7,9) | (1,1,3) |

**Table 9. Criterion constraints.**

| Criterion | Description | Weight constraint | Rank best scale | Rank worst scale |
|---|---|---|---|---|
| $C_1$ | Ease of WSS disassembly | 0.168–0.520 | 1 | 13 |
| $C_2$ | Ease of mesh expansion | 0.164–0.496 | 2 | 10 |
| $C_3$ | Ease of flattening | 0.117–0.354 | 13 | 1 |
| $C_4$ | Cost saving | 0.156–0.462 | 4 | 8 |
| $C_5$ | Sheet steel/weight ratio | 0.151–0.446 | 5 | 9 |
| $C_6$ | Sheet/mesh ratio | 0.157–0.428 | 6 | 12 |
| $C_7$ | TDWP genetic transformation | 0.161–0.496 | 3 | 11 |
| $C_8$ | Eco-cost saving | 0.134–0.360 | 11 | 2 |
| $C_9$ | Pollution reduction | 0.131–0.376 | 12 | 3 |
| $C_{10}$ | Employment | 0.143–0.446 | 7 | 6 |
| $C_{11}$ | Human development | 0.138–0.428 | 8 | 7 |
| $C_{12}$ | Ergonomic | 0.126–0.386 | 9 | 4 |
| $C_{13}$ | Workers' health | 0.121–0.376 | 10 | 5 |

The statistical modelling of sustainability gave values of $C = 0.95$, $T = 0.9$ and $E = 0.85$, while the fuzzy analytical hierarchy process assessment refers to low SI due to weakness of communications during the development phase of exterior components, giving $S = 0.68$ and $M = 0.4$.

Subsequently, scenario-based analysis was applied to propose regulating equations for multiple-bottom-line evaluation. Here, $T$, $C$, and $E$ were used to evaluate the remanufacturability, while the $M$ and $S$ values were also added to evaluate the overall sustainability. Such regulating equations could help developers and policy-makers decide which factors need to be emphasised, the intended SI to be reached is:

$$SI = [W_M M^5 + W_T T^4 + W_E E^x + W_S S^x + W_C C^x] \tag{6}$$

The $R^2$ values from Figs 5–7 were used as the mean SI values for the exterior component designs within the same SI range, where the lower, middle, and upper ranges had values of 0.4, 0.68, and 0.93, respectively, while the integrated SI value obtained from the average of the statistical modelling results was 0.88. These values were then used to develop the following scenario-based analysis to calculate normalised SI values. The order of importance of the indices for each scenario is defined in Table 12.

According to first scenario, social development is the most important, the economic development is the second-most important, while environmentally conscious development will be satisfied when the other thresholds are achieved. The SI for this scenario ($SI_1$) was calculated

**Table 10. Car part alternative constraints.**

| Criterion | Description | Weight constraint | Rank best scale | Rank worst scale |
|---|---|---|---|---|
| $A_1$ | Roof | 0.168–0.520 | 1 | 7 |
| $A_2$ | Hood | 0.164–0.496 | 2 | 6 |
| $A_3$ | Boot | 0.164–0.478 | 3 | 5 |
| $A_4$ | Front door | 0.138–0.386 | 4 | 4 |
| $A_5$ | Rear door | 0.134–0.376 | 5 | 3 |
| $A_6$ | Rear fender | 0.121–0.360 | 6 | 2 |
| $A_7$ | Front fender | 0.117–0.354 | 7 | 1 |

**Table 11. Topological discontinuity constraints.**

| Criterion | Description | Weight constraint |
|---|---|---|
| P | Prominence | $0.039 \leq P \leq 0.462$ |
| $P_r$ | Protrusion | $0.046 \leq P_r \leq 0.140$ |
| I | Isolation | $0.039 \leq I \leq 0.496$ |
| $P_o$ | Pocket | $0.039 \leq P_o \leq 0.194$ |
| G | Groove | $0.039 \leq G \leq 0.241$ |
| D | Dent | $0.045 \leq D \leq 0.496$ |

as follows, giving a value of 0.46.

$$SI_1 = \frac{[W_M M^5 + W_T T^4 + W_E E^3 + W_C C^2 + W_S S^1]}{5} \tag{7}$$

According to second scenario, social development is the most important, environmentally conscious development is the second-most important, while the economic development will be satisfied when the other thresholds are achieved. The SI for this scenario ($SI_2$) was calculated as follows, giving a value of 0.46.

$$SI_2 = \frac{[W_M M^5 + W_T T^4 + W_E E^3 + W_C C^2 + W_S S^1]}{5} \tag{8}$$

According to third scenario, economic development is the most important, social development is the second most important while environment conscious development will be satisfied when the other thresholds are achieved. The SI for this scenario ($SI_3$) was calculated as follows, giving a value of 0.47.

$$SI_3 = \frac{[W_M M^5 + W_T T^4 + W_E E^3 + W_C C^2 + W_S S^1]}{5} \tag{9}$$

According to fourth scenario, economic development is the most important, environmentally conscious development is the second-most important, while social development will be satisfied when the other thresholds are achieved. The SI for this scenario ($SI_4$) was calculated as follows, giving a value of 0.49.

$$SI_4 = \frac{[W_M M^5 + W_T T^4 + W_E E^3 + W_C C^2 + W_S S^1]}{5} \tag{10}$$

According to fifth scenario, environmentally conscious development is the most important, economic development is the second-most important, while social development will be satisfied when the other thresholds are achieved. The SI for this scenario ($SI_5$) was calculated as

**Table 12. Scenario based analysis.**

| Importance | First scenario | Second scenario | Third scenario | Fourth scenario | Fifth scenario | Sixth scenario |
|---|---|---|---|---|---|---|
| Most important | S | S | C | C | E | E |
| 2nd-most important | C | E | S | E | C | S |
| Will be satisfied when others are achieved | E | C | E | S | S | C |
| Easy to satisfy | T | T | T | T | T | T |
| Least important | M | M | M | M | M | M |

follows, giving a value of 0.46.

$$\text{SI}_5 = \frac{[W_M M^5 + W_T T^4 + W_E E^3 + W_C C^2 + W_S S^1]}{5} \tag{11}$$

According to sixth scenario, environmentally conscious development is the most important, social development is the second most important, while economic development will be satisfied when the other thresholds are achieved. The SI for this scenario ($\text{SI}_6$) was calculated as follows, giving a value of 0.45.

$$\text{SI}_6 = \frac{[W_M M^5 + W_T T^4 + W_E E^3 + W_C C^2 + W_S S^1]}{5} \tag{12}$$

Since high sustainability can be obtained by remanufacturing WSS into MSS, technical development is easy to achieve. Furthermore, as remanufacturing can be started in the form of a small-to-medium business by a few individuals, management development is the least important as entering the market of remanufactured MSS could be achieved without deep management experience.

Analysis of the WSS-MSS remanufacturing of exterior components from cars manufactured in the period of 2009–2019 gave overall sustainability indices of 0.45–049, which are considered quite low. Hence, extensive efforts are required to carefully plan the remanufacturing potential of the exterior panels during the design phase to develop a sustainable business model. In summary, 25% of the analysed designs had good potential to be remanufactured sustainably, while 55% of the designs had SI values lower than the sustainability threshold, which would require a high degree of modification to achieve sustainability. Finally, 20% of the designs had SI values lower than the threshold, and are hence, unsustainable to remanufacture.

By using scenario-based analysis, a normalisation process was applied to homogenise the SI values and reduce them to six equations. The weights of the individual economic, environmental, social, technical, and management indices were multiplied by the weights of importance of the individual sustainability indices.

At the beginning of the analysis, 20 SI values were output based on the fuzzy analytical hierarchy of the sustainability assessment of 200 car designs. Pre-plotting of 20 SI values enabled them to be divided into three groups, where the highest SI values indicated the designs that are most suitable for sustainable remanufacturing. The designs with moderate SI values could be moved up to pass the sustainability threshold or move down toward unsustainability depending on the strength of integration of social development enablers. The designs with the lower SI values were clearly unsustainable for remanufacturing due to weakness in the integration of the management enablers, and these designs should be avoided in remanufacturing processes.

## Conclusions

The suitability of different designs of exterior components of cars manufactured during 2009–2019 for either remanufacturing into MSS or traditional recycling was successfully evaluated using thirteen criteria. The proposed methodology was considered effective for assessing the sustainability of WSS–MSS remanufacturing. The statistical modelling was based on resource and equipment allocation, and global policies aiming to close the remanufacturing supply chain loop. The statistical modelling was a macro-scale analysis, where differences in the ease of disassembly of the exterior components were observed. However, fuzzy modelling analysis showed that the roof was the easiest, and the doors were the hardest, to remanufacture. The ease of mesh expanding was highly sensitive to topological discontinuity constraints according to fuzzy modelling, while statistical modelling did not consider these constraints. Both fuzzy

and statistical modelling assumed that the ease of flattening was not affected by the type of exterior component since it was transformed into MSS. The cost-saving feasibility calculated statistically was higher than that calculated using fuzzy analysis, since the remanufacturing equipment should have the same allocation ability and productivity worldwide. The results of fuzzy modelling showed that the WSS/weight ratio, WSS/MSS ratio, TDWP genetic transformation, eco-cost saving, pollution reduction feasibility, employment, human development, ergonomic, and workers health were highly important criteria, while the statistical modelling did not account for such detailed criteria. The statistical modelling was based on theoretical designs of exterior components which are simple and have high continuity; therefore, the high feasibilities were expected. In contrast, fuzzy modelling was based on individual differences of recent exterior component designs, which resulted in a wide variation in SI values.

The statistical analysis resulted in a sufficiently high SI value of 0.88 (including $T$, $C$, and $E$ indices), while fuzzy modelling gave much lower SI values of 0.024–0.535. Even the highest value of ~0.5 indicates that significant efforts are required to reach a high sustainability, i.e., SI close to 1. The SI value of 0.93 for the high region of the SI curve was attributed to good communication and management experience in social development during the design phase of the exterior components, while the value of 0.68 for the middle region was due to weak social development, and the value of SI of 0.4 for the low region was due to weak management practises.

Plotting the SI values was a successful method to estimate the important indicators, and showed that social and management factors are of secondary importance for manufacturing–remanufacturing business developers. The scenario-based analysis highlighted six scenarios of multiple bottom line sustainability, and six modelling equations were provided for the passenger car designs. Evaluation of the 200 passenger car designs showed that the new generations of cars continue to have a low SI due to high degree of complexity of the car bodies and the associated complex deformation of the formed metal sheets which makes them harder to remanufacture. Hence, to achieve a sustainable manufacturing–remanufacturing closed loop in the automotive industry, sustainable design of the metal shell components is required, considering the production of steel sheets free of TDWPs, while maintaining the lightweight structure.

## Author Contributions

**Conceptualization:** Ziyad Tariq Abdullah.

**Data curation:** Ziyad Tariq Abdullah.

**Formal analysis:** Ziyad Tariq Abdullah.

**Writing – original draft:** Ziyad Tariq Abdullah.

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
