## [Decision Letter · Decision Letter 0]

15 Jul 2021

PONE-D-21-18014

Remanufacturing End-of-life Passenger Car Waste Sheet Steel into Mesh Sheet: A Sustainability Assessment

PLOS ONE

Dear Dr. Tariq,

Thank you for submitting your manuscript to PLOS ONE. After careful consideration, we feel that it has merit but does not fully meet PLOS ONE’s publication criteria as it currently stands. Therefore, we invite you to submit a revised version of the manuscript that addresses the points raised during the review process.

We look forward to receiving your revised manuscript.

Kind regards,

Dragan Pamucar

Academic Editor

PLOS ONE

Journal Requirements:

3. Please upload a copy of Figure 1, 2, 3, 4, 5, 6 and 7 to which you refer in your text on page 6, 10, 19, 20, 21, 22 and 23. If the figure is no longer to be included as part of the submission please remove all reference to it within the text.

Reviewers' comments:

Reviewer's Responses to Questions

**Comments to the Author**

1. Is the manuscript technically sound, and do the data support the conclusions?

Reviewer #1: Yes

Reviewer #2: Yes

2. Has the statistical analysis been performed appropriately and rigorously? 

Reviewer #1: Yes

Reviewer #2: Yes

3. Have the authors made all data underlying the findings in their manuscript fully available?

Reviewer #1: Yes

Reviewer #2: Yes

4. Is the manuscript presented in an intelligible fashion and written in standard English?

Reviewer #1: Yes

Reviewer #2: Yes

5. Review Comments to the Author

Reviewer #1: The paper "Remanufacturing End-of-life Passenger Car Waste Sheet Steel into Mesh Sheet: A Sustainability Assessment" falls within the scope of the journal "Plos One" but doesn't meet the standard quality of the paper that should be published in one prestigious journal in the current version. A lot of core elements of one well-written and performed study is missing, so the paper needs major improvements.

Please consider the following comments.

- Clear aims, the main contributions, and novelty are missing in the abstract. The abstract should be concise with a description of the core elements of the paper.

- The paper isn't well structured. Should be restructured: 1. Introduction, 2. Literature review, 3. Methodology, 4. Results or case study with subsections, 5. Discussion, 6 Conclusion

- In the introduction section the following tasks should be fulfilled: the introduction should give an overview of the field significance, and should consider the following main questions: What are the gaps in literature? What are the main aims of this article?"

- Most of the current section Introduction should be moved to new-formed 2. Literature review.

- In the Methodology section should be ensured the diagram flow with showed research.

- Why you have decided to use best-worst multi-criteria analysis. Explain. Also, should be represented each step with equations.

- More relevant references newer date should be added:

1) Biswas, T. K., & Das, M. C. (2020). Selection of the barriers of supply chain management in Indian manufacturing sectors due to COVID-19 impacts. Operational Research in Engineering Sciences: Theory and Applications, 3(3), 1-12.

2) Fazlollahtabar, H., & Kazemitash, N. (2021). Green supplier selection based on the information system performance evaluation using the integrated Best-Worst Method. Facta Universitatis, Series: Mechanical Engineering.

3) Vasiljević, M., Fazlollahtabar, H., Stević, Ž., & Vesković, S. (2018). A rough multicriteria approach for evaluation of the supplier criteria in automotive industry. Decision Making: Applications in Management and Engineering, 1(1), 82-96.

- Comparative analysis with other MCDM methods is missing.

Reviewer #2: Thanks for giving me opportunity to review this article.

topic is very novel, different and attracting for readers. abstract and introduction sections are well designed and written. some recent references can be added in terms of WSS-MSS remanufacturing process. methodology section is explicitly written and stated too.

Under results section there needs to be some changes.

Equation 8 needs to be rewritten as S12=(WMM5+WTT4+WCC3+WEE2+WSS1)/5

Same changes need to be done for Eqs. (9-12).

some different future suggestions can be added to conclusion section.

there are some typos so proofreading can be useful.

6. PLOS authors have the option to publish the peer review history of their article (what does this mean?). If published, this will include your full peer review and any attached files.

Reviewer #1: No

Reviewer #2: No

---

## [Author Response · Author response to Decision Letter 0]

30 Jul 2021

Reviewer #1: The paper "Remanufacturing End-of-life Passenger Car Waste Sheet Steel into Mesh Sheet: A Sustainability Assessment" falls within the scope of the journal "Plos One" but doesn't meet the standard quality of the paper that should be published in one prestigious journal in the current version. A lot of core elements of one well-written and performed study is missing, so the paper needs major improvements.

Thank you for the opportunity to revise the paper. I have included most of your suggestions, as described here.

Please consider the following comments.

- Clear aims, the main contributions, and novelty are missing in the abstract. The abstract should be concise with a description of the core elements of the paper.

The abstract has been modified to include these parts (Page 2, lines 15-18, 30-35)

- The paper isn't well structured. Should be restructured: 1. Introduction, 2. Literature review, 3. Methodology, 4. Results or case study with subsections, 5. Discussion, 6 Conclusion

While I did not follow the suggested section titles, the structure of the submitted paper did indeed follow the suggested format. To further follow this standard format, the second section has been renamed “Literature Review,” and “Modelling methods” was changed to “Methodology.” As the Results and Discussion section is quite short, I believe that a combined section is acceptable. 

- In the introduction section the following tasks should be fulfilled: the introduction should give an overview of the field significance, and should consider the following main questions: What are the gaps in literature? What are the main aims of this article?"

Field significance: waste vehicles account for a large fraction of land fill and contain high embodied energy that has the potential to be recovered. Remanufacturing processes have the potential to recover this “waste” material and can provide a profitable business strategy. This was addressed in the original document (Introduction). 

Gaps in the knowledge: There are currently few practical remanufacturing methods for waste sheet steel. Most is sent to land fill or recovered by energy-intensive smelting processes. The proposed remanufacturing process to produce mesh sheet has potential to address these problems; however, its viability and sustainability needs to be clarified. Some further discussion of this gap in the knowledge has been added to the Introduction (Page 5-6, lines 112-113).

Main aims: The aim of this study was to evaluate the sustainability of the proposed remanufacturing process. This was stated in the final paragraph of the introduction. 

- Most of the current section Introduction should be moved to new-formed 2. Literature review.

I respectfully disagree with this suggestion. In its revised form, I believe that the Introduction follows the PLOS ONE guidelines and does not include an extensive discussion of the literature. Perhaps the reviewer is referring to the section named “WSS-MSS Remanufacturing”. This is a separate section to the Introduction, and while this part does indeed discuss existing literature related to the potential waste stream, it is introducing the proposed WSS-MSS process, which is a novel process proposed in this paper. Hence, I do not think that it is appropriate to include this as a literature review.

- In the Methodology section should be ensured the diagram flow with showed research.

Thank you for this good suggestion. A flowchart of the analysis method was included (Page 12, line 240)

- Why you have decided to use best-worst multi-criteria analysis. Explain. Also, should be represented each step with equations.

The best-worst multi-criteria method was chosen as such techniques can provide a more integrated assessment to reduce large fluctuations in weight values. This explanation has been included in the text (Page 20–21; Line 365-378). 

The relevant equations have been added as Eq. (6)–(9) (Pages 19–20; lines 346-364).

- More relevant references newer date should be added:

1) Biswas, T. K., & Das, M. C. (2020). Selection of the barriers of supply chain management in Indian manufacturing sectors due to COVID-19 impacts. Operational Research in Engineering Sciences: Theory and Applications, 3(3), 1-12.

2) Fazlollahtabar, H., & Kazemitash, N. (2021). Green supplier selection based on the information system performance evaluation using the integrated Best-Worst Method. Facta Universitatis, Series: Mechanical Engineering.

3) Vasiljević, M., Fazlollahtabar, H., Stević, Ž., & Vesković, S. (2018). A rough multicriteria approach for evaluation of the supplier criteria in automotive industry. Decision Making: Applications in Management and Engineering, 1(1), 82-96.

Thank you for these suggestions, they have been added as references [29], [30], and [31] in the revised manuscript.

- Comparative analysis with other MCDM methods is missing.

A comparison has been added to the text, as required (page 20,21, lines 365–378).

Reviewer #2: Thanks for giving me opportunity to review this article.

topic is very novel, different and attracting for readers. abstract and introduction sections are well designed and written. methodology section is explicitly written and stated too.

Thank you for the comments. I am happy that the reviewer was able to follow the argument of the manuscript.

some recent references can be added in terms of WSS-MSS remanufacturing process. 

As this is a newly proposed remanufacturing process, there is little previous literature. I have added two of my previous conference papers to the original manuscript (refs [13,14]), but to the best of my knowledge, there are no recent references. Other references in the field of machine tool remanufacturing were included as refs [17-20].

Under results section there needs to be some changes.

Equation 8 needs to be rewritten as S12=(WMM5+WTT4+WCC3+WEE2+WSS1)/5

Same changes need to be done for Eqs. (9-12).

Thank you for bringing this to my attention. I have now defined a more general equation (Eq. 10; page 25, line 457) and this was used to calculate the SI values for all scenarios, where the exponent for E, S, and C varied. These changes have been made (pages 26-27, lines 469, 473, 478, 482, 487, 491) 

some different future suggestions can be added to conclusion section.

Thank you for this suggestion, I have included some future work (page 30, 560-563)

there are some typos so proofreading can be useful.

The revised manuscript has been thoroughly checked before resubmission.

---

## [Decision Letter · Decision Letter 1]

27 Sep 2021

Remanufacturing End-of-life Passenger Car Waste Sheet Steel into Mesh Sheet: A Sustainability Assessment

PONE-D-21-18014R1

Dear Dr. Tariq,

We’re pleased to inform you that your manuscript has been judged scientifically suitable for publication and will be formally accepted for publication once it meets all outstanding technical requirements.

Kind regards,

Dragan Pamucar

Academic Editor

PLOS ONE

Additional Editor Comments (optional):

Reviewers' comments:

Reviewer's Responses to Questions

**Comments to the Author**

1. If the authors have adequately addressed your comments raised in a previous round of review and you feel that this manuscript is now acceptable for publication, you may indicate that here to bypass the “Comments to the Author” section, enter your conflict of interest statement in the “Confidential to Editor” section, and submit your "Accept" recommendation.

Reviewer #1: All comments have been addressed

Reviewer #2: All comments have been addressed

2. Is the manuscript technically sound, and do the data support the conclusions?

Reviewer #1: Yes

Reviewer #2: (No Response)

3. Has the statistical analysis been performed appropriately and rigorously? 

Reviewer #1: Yes

Reviewer #2: (No Response)

4. Have the authors made all data underlying the findings in their manuscript fully available?

Reviewer #1: Yes

Reviewer #2: (No Response)

5. Is the manuscript presented in an intelligible fashion and written in standard English?

Reviewer #1: Yes

Reviewer #2: (No Response)

6. Review Comments to the Author

Reviewer #1: The authors have revised the paper and now is more quality. Also, the authors have addressed all my comments.

Reviewer #2: All necessary revisions are made by author(s). So manuscript is suitable for publication in PLOS ONE journal.

7. PLOS authors have the option to publish the peer review history of their article (what does this mean?). If published, this will include your full peer review and any attached files.

Reviewer #1: No

Reviewer #2: No

---

## [Editor Report · Acceptance letter]

1 Oct 2021

PONE-D-21-18014R1 

Remanufacturing end-of-life passenger car waste sheet steel into mesh sheet: A sustainability assessment 

Dear Dr. Abdullah:

I'm pleased to inform you that your manuscript has been deemed suitable for publication in PLOS ONE. Congratulations! Your manuscript is now with our production department. 

Kind regards, 

on behalf of

Dr. Dragan Pamucar 

Academic Editor

PLOS ONE